# Effects of a Maximal Exercise Followed by a Submaximal Exercise Performed in Normobaric Hypoxia (2500 m), on Blood Rheology, Red Blood Cell Senescence, and Coagulation in Well-Trained Cyclists

**DOI:** 10.3390/metabo13020179

**Published:** 2023-01-25

**Authors:** Romain Carin, Gabriel Deglicourt, Hamdi Rezigue, Marie Martin, Christophe Nougier, Camille Boisson, Yesim Dargaud, Philippe Joly, Céline Renoux, Philippe Connes, Emeric Stauffer, Elie Nader

**Affiliations:** 1Laboratoire Interuniversitaire de Biologie de la Motricité (LIBM) EA7424, Team “Vascular Biology and Red Blood Cell”, Universié Claude Bernard Lyon 1, Université de Lyon, 69007 Lyon, France; 2Laboratoire d’Excellence du Globule Rouge (Labex GR-Ex), PRES Sorbonne, 79015 Paris, France; 3Exploration Fonctionnelle Respiratoire, Médecine du Sport et de l’activité Physique, Hospices Civils de Lyon, Hôpital de la Croix Rousse, 69004 Lyon, France; 4Service d’hématologie-hémostase, Hospices Civils de Lyon, 69002 Bron, France; 5EA 4609-Hémostase et Thrombose, SFR Lyon Est, Université Claude Bernard Lyon I, 69100 Lyon, France; 6Service de Biochimie et de Biologie Moléculaire, Centre de Biologie et de Pathologie Est, Hospices Civils de Lyon, 69002 Bron, France

**Keywords:** hemorheology, endurance, hemostasis, altitude, eryptosis, cycling

## Abstract

Acute normoxic exercise impacts the rheological properties of red blood cells (RBC) and their senescence state; however, there is a lack of data on the effects of exercise performed in hypoxia on RBC properties. This crossover study compared the effects of acute hypoxia vs. normoxia on blood rheology, RBC senescence, and coagulation during exercise. Nine trained male cyclists completed both a session in normoxia (FiO_2_ = 21%) and hypoxia (FiO_2_ = 15.3% ≈ 2500 m). The two sessions were randomly performed, separated by one week, and consisted of an incremental and maximal exercise followed by a 20 min exercise at the first ventilatory threshold (VT1) on a home-trainer. Blood samples were taken before and after exercise to analyze hematological parameters, blood rheology (hematocrit, blood viscosity, RBC deformability and aggregation), RBC senescence markers (phosphatidylserine (PS) and CD47 exposure, intraerythrocyte reactive oxygen species (ROS), and calcium content), and blood clot viscoelastic properties. Hemoglobin oxygen saturation (SpO_2_) and blood lactate were also measured. In both conditions, exercise induced an increase in blood viscosity, hematocrit, intraerythrocyte calcium and ROS content, and blood lactate concentration. We also observed an increase in blood clot amplitude, and a significant drop in SpO_2_ during exercise in the two conditions. RBC aggregation and CD47 exposure were not modified. Exercise in hypoxia induced a slight decrease in RBC deformability which could be related to the slight increase in mean corpuscular hemoglobin concentration (MCHC). However, the values of RBC deformability and MCHC after the exercise performed in hypoxia remained in the normal range of values. In conclusion, acute hypoxia does not amplify the RBC and coagulation changes induced by an exercise bout.

## 1. Introduction

Through their effects on the regulation of blood flow and tissue perfusion, the physical and dynamic properties of blood and its constituents (i.e., blood rheology) may impact on endurance performance [1,2]. Red blood cells (RBCs) need to be highly deformable to easily flow through the smallest capillaries (whose diameters are sometimes less than that of the RBC itself) and transport oxygen to the tissues [3,4]. In addition, any loss of RBC deformability or any increase in RBC aggregation may lead to a rise in blood viscosity which would be accompanied by a rise of vascular resistance [5].

Due to the important role played by blood rheology on tissue perfusion and possibly on endurance performance [1,2], several studies have investigated the effect of an acute cycling maximal and/or submaximal exercise on blood rheology. Most of the studies have reported an increase in blood viscosity at the end of the exercise, mainly in relation with the rise in hematocrit subsequent to water loss [6,7], a fluid shift from the vascular to the extravascular compartment, or even the release of stored RBCs into the spleen following adrenergic-related stress [8,9]. The changes in RBC deformability and RBC aggregation properties during submaximal and maximal cycling exercises may also participate in the changes in blood viscosity [8,10,11,12]. It has been observed that a decrease in RBC deformability during a short and intense cycling exercise resulted in a rise of blood viscosity [10].

In vitro studies have reported that several physiological factors modulated by exercise can affect the RBC rheological properties as well as their senescence state. The increase in blood lactate concentration and the decrease in pH can lead to activation of nonselective cation channels located on the membrane of RBCs, leading to RBC dehydration, and, therefore, reducing their deformability [13,14]. Similarly, through lipid and protein oxidation, accumulation of reactive oxygen species (ROS) can damage RBCs, leading to a decrease in RBC deformability and an increase in the strength of RBC aggregates [15,16]. Oxidative stress is also involved in triggering senescence [17]. RBC senescence is characterized, among other things, by a calcium (Ca^2+^) entry into RBCs, leading to RBC shrinkage and breakdown of the cell membrane asymmetry with translocation of phosphatidylserine (PS) from the inner leaflet of the cell membrane to the RBC surface [18]. However, studies on the effects of exercise on RBC senescence markers are rather scarce and have been conducted mainly during running exercises [19,20]. The externalization of PS may also promote the activation of coagulation through the formation of a tenase and prothrombinase complex [21,22] and could explain the findings reported in some studies showing a shortening of blood clot formation time during exercise using rotational thromboelastography [23].

The altitude environment is becoming increasingly common among endurance athletes to optimize their performance following the stimulation of erythropoiesis. Hypoxic conditions (simulated or real) are characterized by a drop in inspiratory oxygen pressure, leading to a decrease in O_2_ binding to hemoglobin and thus a drop in arterial O_2_ saturation [24]. Acute hypoxia has been found to promote systemic oxidative stress in humans [25,26]. During exercise in hypoxia, the reduction in arterial O_2_ desaturation was reported to positively correlate with plasma markers of oxidative stress [27]. In nonacclimatized to altitude athletes, performing a maximal and/or a submaximal exercise in hypoxia results in a greater increase in blood lactate concentration compared to the same exercise performed at sea level [28,29]. In vitro hypoxic studies on RBC reported a decrease in RBC deformability (FiO_2_ = 0%, time of exposure: 60 min) [30] and a rise in PS externalization (FiO_2_ = 5%, time of exposure: 24 h) [31,32] after a severe hypoxic exposure. One could, thus, expect greater changes in RBC rheology and senescence markers in athletes exercising in hypoxia compared to normoxic conditions. Moreover, an increase in prothrombin expression after exposure to hypobaric hypoxia (30 min at 2400 m) has been observed [33]. Similarly, one could suspect additional effects of hypoxia and exercise on the formation of clot.

The aim of this study was to compare the effects of an exercise conducted in normobaric hypoxia (FiO_2_ = 15.3% ≈ 2500 m) vs. normoxia on blood rheology, RBC senescence and blood coagulation in endurance-trained cyclists. We hypothesized that the exercise performed in hypoxia would result in greater changes on those parameters compared to the same exercise conducted in normoxia. 

## 2. Materials and Methods

### 2.1. Subjects

Nine endurance-trained male cyclists (30.0 ± 8.5 years; 66.2 ± 7.5 kg; 177.0 ± 2.8 cm) voluntarily took part in this study. After being informed of the experimental procedure and the possible risks associated with the experiment, all participants signed a consent form. The experimental protocol was approved by the local ethics committee (Lyon, France, L16-47), and all the procedures performed during this study were in agreement with the Declaration of Helsinki. Participants practiced cycling for at least 5 years and trained at least three times a week. Subjects were nonsmokers, presented no known cardiovascular, metabolic, or pulmonary pathology, and had not been exposed to an altitude environment (real or simulated) in the three months preceding the intervention. 

### 2.2. Protocol

This crossover study was conducted in a laboratory at sea level (Respiratory Functional Exploration Service, Hospices Civils de Lyon, Croix Rousse Hospital, Lyon, France). Each participant randomly performed a session in normoxia (FiO_2_ = 21%) and a session in normobaric hypoxia (FiO_2_ = 15.3% ≈ 2500 m). Normobaric hypoxia was generated by adding nitrogen (N2) to the inspired air, thus decreasing the fraction of oxygen in the ambient air (Alti-Trainer200, Sport and Medical Technology). The participants performed two sessions separated by one week. Tests were executed at the same time of day to avoid the effects of circadian rhythm. During each session, participants performed a maximal incremental test followed by a 20 min submaximal exercise at the first ventilatory threshold (VT1). The tests were carried out on a home-trainer (Saris H3 direct drive), in a laboratory where the temperature was kept constant between 20 and 25 °C. Athletes were asked to refrain from any physical activity the day before the tests. Subjects were weighted before and after exercise. Water intake was not allowed during the tests. Venous blood samples were drawn in a sitting position from the antecubital vein, at rest before the maximal incremental test (Pre) and 3 min after the end of the submaximal exercise at VT1 (Post) in EDTA or citrate tubes (Vacutainer, Becton Dickinson, Rutherford, NJ, USA), for blood rheological, RBC senescence, hematological, and coagulation parameters.

### 2.3. Maximal Incremental Test

In order to determine the maximal oxygen consumption (VO_2max_), the maximal aerobic power (MAP), and the first ventilatory threshold (VT1) during each of the two sessions, the athletes first performed a maximal incremental test on the home-trainer. This test started with a 3 min warm up at 90 watts. The pedaling frequency had to be maintained between 70 and 90 RPM. The power was increased by 30 watts per minute, until exhaustion. Gas exchanges and ventilation were measured using a breath-by-breath automated exercise metabolic system (COSMED, Rome, Italy), and heart rate monitoring was performed by electrocardiography. VO_2max_ was considered when the usual criteria were met (i.e., VO_2_ plateau despite an increase in the intensity of the effort, heart rate close to the theoretical maximum heart rate (220-age +/−10%), and respiratory exchange ratio higher than 1.1) [34]. The first ventilatory threshold (VT1) was calculated using the methods of Wasserman et al. [35] and Beaver et al. (V-slope method [36]). A 6 min active recovery period at 90 watts was then respected for all athletes. 

### 2.4. Submaximal Exercise at VT1

After the recovery period, the athletes performed a 20 min submaximal exercise at VT1. The pedaling frequency had to be maintained between 70 and 90 RPM. Gas exchanges, ventilation, and heart rate were measured in the same way as during the incremental test.

### 2.5. Hemoglobin Saturation

During both sessions, the noninvasive pulse ear oximetry (Medisoft, Sorinnes, Belgium) method was used to assess hemoglobin saturation (SpO_2_) during the maximal incremental test and the submaximal exercise at VT1. This method has been proven to be valid and reliable for measurement of significant falls in SpO_2_ during exercise [37]. A significant desaturation was considered when the fall in SpO_2_ was ≥4% compared to resting condition [38].

### 2.6. Hematological Parameters, Fibrinogen, Lactate, and Glucose

Hematological parameters were measured using an hematological analyzer (Excell 2280, Drew Scientific, Miami Lakes, FL, USA). Analyses were performed within one hour after blood sampling to avoid RBC deterioration. Fibrinogen was measured using a standard coagulation method (ACLTOP750, Werfen). Blood lactate and glucose concentrations were analyzed (Pre and Post) with a drop of blood collected at the finger level with a lactate meter (Nova Biomedical, Cheshire, UK) and a glucometer (Abbot Diabetes Care, Oxon, UK), respectively.

### 2.7. Blood Rheological Parameters

#### 2.7.1. Blood Viscosity and Hematocrit

Blood viscosity was measured after complete blood oxygenation, at native hematocrit (Hct), 25 °C and several shear rates (11.5; 22.5; 45; 90; 225 s^−1^) using a cone/plate viscometer (Brookfield DVII+ with CPE40 spindle, Brookfield Engineering Labs, Natick, MA, USA), as recommended [39]. Hematocrit was determined by the micromethod after blood microcentrifugation at 1500 g for five minutes at 20 °C (Pico 17, Thermo Scientific, Illkirch, France).

#### 2.7.2. Red Blood Cell Deformability

Red blood cell deformability under isotonic condition was assessed at 37 °C and at several shear stresses (from 0.3 to 30 Pa) by laser diffraction analysis (ektacytometry), using the laser-assisted optical rotational cell analyzer (LORRCA MaxSis, RR Mechatronics, Hoorn, The Netherlands). The system has been described elsewhere in detail [39]. Briefly, 5 μL of blood were mixed with 1 mL polyvinylpyrrolidone (PVP; viscosity = 27.1 cP). The blood suspension was placed into a Couette system and increasing shear stresses were applied on the blood suspension. A laser beam was projected from the stationary cylinder through the RBC suspension, and the resulting diffraction pattern was captured by a video camera and analyzed by a computer in order to calculate an elongation index (EI), which reflects RBC deformability. The maximum elongation index (EImax) and the shear stress required to deform RBCs by half (SS1/2) were determined by a Lineviewer Burk model [40]. 

#### 2.7.3. Red Blood Cell Aggregation 

Red blood cell aggregation properties were determined at 37 °C and at a standardized Hct (i.e., 40%) and after complete oxygenation of the blood, by syllectometry (changes in backscattered light intensity over time) using the laser-assisted optical rotational cell analyzer (LORRCA MaxSis, RR Mechatronics, Hoorn, The Netherlands). This method has been described in detail by Baskurt et al. [39]. The minimum shear rate required to break the RBC aggregates formed (γmin; RBC aggregates strength) was determined by an iteration procedure [39].

### 2.8. Red blood Cell Senescence Assessment

#### 2.8.1. RBCs Preparation

Blood collected in citrate tubes was centrifuged (1000 g, 10 min at 20 °C) and plasma and buffy coat were discarded. RBCs were washed in PBS, and then resuspended at 0.4% Hct in PBS buffer containing 2.5 mM Ca^2+^.

#### 2.8.2. Phosphatidylserine (PS) Exposure

PS exposure on the outer membrane leaflet of RBCs was evaluated by Annexin V-FITC binding to this phospholipid. RBCs suspensions were incubated, protected from light, 30 min at 37 °C with Annexin V-FITC (1:200 dilution, Beckman Coulter, Pasadena, CA, USA). Immediately after incubation, samples were diluted and analyzed by FACS (BD Accuri C6, Franklin Lakes, NJ, USA). PS exposure was measured in the FITC channel (with an excitation wavelength of 488 nm and an emission wavelength of 530 nm) according to the manufacturer’s instructions. For each sample, 50,000 events gated for the appropriate FSC were counted. PS externalization was assessed by the percentage of Annexin V FITC-positive RBCs.

#### 2.8.3. Intracellular Reactive Oxygen Species (ROS)

Intracellular RBC ROS was determined using 2′,7′–dichlorofluorescin diacetate (DCFDA, Sigma-Aldrich, Saint-Quentin-Fallavier, France). RBC suspensions at 0.4% Hct were incubated 30 min at 37 °C in the dark with 20 µM of DCFDA (Sigma-Aldrich, Saint-Quentin-Fallavier, France). Immediately after incubation, samples were diluted and analyzed by FACS (BD Accuri C6, Franklin Lakes, NJ, USA) according to the manufacturer’s instructions. Median fluorescence intensity (MFI) of the 50,000 gated events was recorded to quantify ROS levels.

#### 2.8.4. Intracellular Calcium (Ca^2+^)

RBC Ca^2+^ content was measured with Fluo3/AM (Biotium, Fremont, CA, USA) probe. RBC suspensions were incubated 30 min at 37 °C with 5 µM of Fluo3/AM. Immediately after incubation, samples were diluted and analyzed by FACS (BD Accuri C6, Franklin Lakes, NJ, USA) according to the manufacturer’s instructions. MFI of the 50,000 gated events was recorded to quantify Ca^2+^ levels.

#### 2.8.5. CD47 Exposure

CD47 (anti-erythrophagocytosis protein) membrane exposure was assessed by incubating RBC suspensions for 30 min at 37 °C with anti-CD47 antibody (1:34 dilution, Miltenyi, Cologne, Germany). Immediately after incubation, samples were diluted and analyzed by FACS (BD Accuri C6, Franklin Lakes, NJ, USA) according to the manufacturer’s instructions. MFI of the 50,000 gated events was recorded to quantify CD47 exposure levels. 

### 2.9. Rotational Thromboelastometry

Coagulation activation and blood clot polymerization parameters were determined with rotational thromboelastometry (ROTEM^®^ delta, Werfen, TEM International, Kampala, Germany) in NATEM and EXTEM mode at 37 °C for 45 min. 

For both NATEM and EXTEM mode, samples were recalcified with 0.2 M CaCl_2_^−^ solution. Then, only in the EXTEM mode, tissue factor was added to explore the extrinsic pathway of coagulation. 

Several parameters were analyzed: (1) coagulation time (CT), which corresponds to the time required to reach an amplitude of 2 mm after the beginning of the test; (2) clot formation time (CFT), which corresponds to the time required to reach an amplitude of 20 mm from the amplitude of 2 mm; (3) alpha angle (α), which reflects the kinetics of clot formation and corresponds to the angle between the baseline and a tangent to the clotting curve through the 2 mm point; (4) maximum clot firmness (MCF), in millimeters, which corresponds to the maximum amplitude of clot firmness reached during the test; and (5) the amplitude of clot firmness 5, 10, 20, and 30 min after CT (A5, A10, A20, and A30, respectively).

### 2.10. Statistical Analysis

Statistical analyses were performed using GraphPad Prism 8 software. Data are expressed as the mean ± standard deviation. For each variable, descriptive analyses were first carried out, then the distribution normality and the variances’ homogeneity were verified. The effect of normoxia vs. hypoxia on VO_2max_, VO_2_ and percentage of VO_2_ at VT1, MAP, and power at VT1 was analyzed using a paired student T-test. The effects of exercise and condition (hypoxia vs. normoxia) were analyzed using a two-way repeated measures ANOVA completed by a Tukey post hoc test. A *p*-value < 0.05 was considered as significant.

## 3. Results

### 3.1. Subject Characteristics

Table 1 shows the VO_2max_, MAP, VO_2_ at VT1, VT1 expressed in percentage of VO_2max_, and the power output at VT1 developed by the participants both in normoxia and hypoxia. VO_2max_ (−14.4%), MAP (−10.8%), VO_2_ at VT1 (−20.7%), and power output at VT1 (−22.9%) were lower in hypoxia compared to normoxia. No difference was observed between the two conditions for VT1 when expressed in percentage of VO_2max_. (Table 1).

### 3.2. SpO_2_, Fibrinogen, Blood Lactate and Glucose Concentration, and Weight

While SpO_2_ decreased from 99.1 ± 0.7% to 95.1 ± 4.1% in normoxia, SpO_2_ reached a value of 81.6 ± 5.9% in hypoxia. Fibrinogen concentration was not affected by exercise performed in both conditions (Table 2). Exercise induced a significant increase in blood lactate (+140.9% in normoxia and +229.4% in hypoxia, *p* < 0.01) and a significant weight loss (−1.7% in normoxia and −1.5% in hypoxia, *p* < 0.0001) in the two conditions (Table 2). Blood glucose concentration tended to increase after exercise in the two conditions (+15.4% in normoxia and +10.2% in hypoxia, *p* = 0.07; Table 2).

### 3.3. Hematological Parameters

Hematological parameters are shown in Table 3. Mean corpuscular volume (MCV) and mean corpuscular hemoglobin (MCH) were not affected by exercise performed in both conditions. Hematocrit (Hct; +3.8% in normoxia and +2.6% in hypoxia, *p* < 0.05), red blood cell (RBC; +4.5% in normoxia and +2.2% in hypoxia, *p* < 0.01), hemoglobin (Hb; +4.6% in normoxia and +3.4% in hypoxia, *p* < 0.01), white blood cell (WBC; +28.8% in normoxia and +35.4% in hypoxia, *p* < 0.01), and platelet (PLT; +23.1% in normoxia and +28.9% in hypoxia, *p* < 0.01) counts increased significantly after exercise in the two conditions. Mean corpuscular hemoglobin concentration (MCHC) was not affected by exercise in normoxia, but we observed a slight but significant increase in MCHC after the exercise in hypoxia compared to baseline level (+1.12%, *p* < 0.01).

### 3.4. Blood Rheological Parameters

The blood rheological parameters before and after exercise in both conditions are displayed in Figure 1. Exercise caused a rise in blood viscosity measured at 225 s^−1^ in the two conditions (A; +11.8% in normoxia and + 10.6% in hypoxia, *p* < 0.0001) but also when measured at 11.5 s^−1^ to 90 s^−1^ (data not shown). RBC aggregation index (B) and EImax (D) were not modified by exercise whatever the condition. Similarly, RBC aggregates’ strength (γmin) was not modified by exercise whatever the condition (data not shown). While in normoxia, the SS1/2 was not affected by exercise (C, *p* = 0.76), we observed a significant increase in SS1/2 during exercise in hypoxia compared to baseline (+21%, *p* < 0.05).

### 3.5. RBC Senescence

RBC senescence markers measured before and after exercise in both conditions are shown in Figure 2. Exercise induced a significant increase in RBCs positive for PS (A; +7.4% in normoxia and +107.0% in hypoxia), RBCs Ca^2+^ (C; +8.6% in normoxia and +6.3% in hypoxia), and ROS (D; +11.6% in normoxia and +9.1% in hypoxia) contents in the two environmental conditions (*p* < 0.05). Neither exercise nor the condition affected CD47 expression (B).

### 3.6. Coagulation Measurements

The coagulation parameters measured before (Pre) and after (Post) exercise in both normoxic and hypoxic conditions are presented in Table 4 (NATEM and EXTEM). CT, A5, α, and CFT parameters measured in the NATEM and EXTEM modes were not modified by exercise, whatever the environmental condition. A20 and A30 parameters measured in the NATEM (+3.4% and +3.6%, respectively, in normoxia, and +2.3% and +2.9%, respectively, in hypoxia) and EXTEM (+4.4% and +4.1%, respectively, in normoxia, and +3.2% and +4.3%, respectively, in hypoxia) modes increased after exercise both in normoxia and hypoxia. Exercise significantly increased MCF measured in the NATEM mode (+3.4% in normoxia and +3.1% in hypoxia, *p* < 0.01) and A10 measured in the EXTEM mode (+4.6% in normoxia and +2.1% in hypoxia, *p* < 0.05) both in normoxia and hypoxia.

## 4. Discussion

The aim of this study was to investigate the effects of an incremental and maximal exercise followed by a 20 min submaximal exercise conducted at the 1st ventilatory threshold on blood rheology, RBC senescence and coagulation in endurance-trained cyclists and to compare the responses between two environmental conditions: simulated hypoxia vs. normoxia. The main results showed (1) an increase in hematocrit and blood viscosity after exercise whatever the condition (hypoxia or normoxia), (2) a slight increase in MCHC and a decrease in RBC deformability after the exercise performed in hypoxia only, (3) an increase in RBC with externalized PS and a rise of intracellular Ca^2+^ and ROS content after exercise performed in both normoxia and hypoxia, and (4) an activation of the coagulation pathway exercise in both conditions. 

Contrary to Płoszczyca et al. [29], we did not observe a greater increase in blood lactate concentration in hypoxia than in normoxia. These result discrepancies could be linked to the differences in the type of exercise performed (30 km time trial for [29] vs. 20 min submaximal exercise at fixed intensity in our study) and the level of athletes (VO_2max_ was 11.5% higher in our study). Moreover, the 20 min submaximal exercise was performed at a similar relative intensity in normoxia and in hypoxia (VT1 expressed as a percentage of the VO_2max_ determined in normoxia or hypoxia), which could partly explain why, in our study, blood lactate concentration was not higher in hypoxia than in normoxia.

The increase in blood viscosity measured after cycling exercise is in agreement with previous studies conducted either at sea level or after one or five nights spent at 2400 above sea level [2,6,8,41,42]. This increase in blood viscosity seems to be mainly due to the rise of hematocrit that may be partly due to dehydration, as suggested by the weight loss occurring after exercise compared to pre-exercise levels. While an excessive increase in blood viscosity may lead to a rise in vascular resistance [1,2], the physiological increase in blood viscosity observed in our study could have stimulated nitric oxide production by endothelial cells to promote compensatory vasodilation [43,44,45] and, therefore, improve tissue perfusion. For instance, a positive correlation between the percentage of changes in blood viscosity and the percentage of changes in circulating nitric oxide and the decrease in vascular resistance during exercise has been reported [46].

Blood viscosity can also be affected by the rheological properties of RBCs [5,47,48]. Indeed, at a given hematocrit, any decrease in RBC deformability and/or increase in RBC aggregation would result in a rise of blood viscosity [5]. While an increase in RBC aggregation has recently been reported following an incremental and maximal cycling exercise [6], we did not observe any change in the present study. The differences concerning the training status of the athletes (VO_2max_ 13% higher in our study) or the type of effort carried out could be responsible for this discrepancy. Fibrinogen contributes to the increase in RBC aggregation [49]. However, we did not observe any change in this parameter after exercise, which could also explain the lack of change in RBC aggregation. The decrease in RBC deformability measured after exercise in hypoxia could have participated in the increase in blood viscosity observed in this condition. However, the decrease in RBC deformability in hypoxic condition after exercise was rather small, which could explain why the increase in blood viscosity induced by the exercise in hypoxia was not greater than during normoxia. Further studies are needed to test whether higher hypoxic stress or greater exercise intensity and/or duration would result in similar results.

In normoxia, we did not observe any changes in RBC deformability following exercise. The data reported in the literature are divergent. After a normoxic exercise, some studies have reported an increase in RBC deformability [2,8], others a decrease [8,50] or no modification [6,7,51]. These discrepancies may be related to differences in the training status of the athletes, the type of exercise performed, or the techniques used to measure RBC deformability. In contrast, RBC deformability slightly decreased below resting values following the exercise performed in hypoxia. A previous study also reported a slight decrease in RBC deformability after a 30 min continuous cycling exercise at low intensity performed in normobaric hypoxia (FiO_2_ = 12% ≈ 4460 m) [52]. However, in this study, the effects of exercise on RBC deformability have not been tested in normoxic conditions. Therefore, it is difficult to dissociate the effects of exercise from those of hypoxic stress. Moreover, an in vitro study observed a decrease in RBC deformability after a 60 min exposure to severe hypoxia (FiO_2_ = 0%) [30]. According to the authors, the increased band 3 clustering in hypoxia seems to be responsible for the decrease in RBC deformability. In our study, although the increase in the MCHC during the exercise performed in hypoxia was very slight, it could have participated in the slight decrease in RBC deformability [53,54]. However, the values observed before and after exercise, both in normoxia and hypoxia, were in the normal ranges and do not indicate major RBC dehydration. An important decrease in RBC deformability may decrease tissue perfusion and oxygenation [3]. However, an association between faster oxygen consumption kinetics and slightly decreased RBC deformability in athletes has been reported [55]. It has been proposed that a slight decrease in RBC deformability could increase the transit time of RBC into the capillary networks and reduce the distance between the RBCs and the vessel wall, which could improve tissue extraction of oxygen from blood [56,57]. By this mechanism, the slightly decreased RBC deformability in hypoxia might have enhanced oxygen exchange between RBC and peripheral tissue, to compensate for the effects of hypoxia on reduced oxygen content, delivery, and consumption.

Exercise was accompanied by an increase in the intracellular Ca^2+^ content, both in normoxia and hypoxia. The activation of Ca^2+^ permeable unselective cation channels caused by the increase in lactate and ROS levels could have been responsible for the massive influx of Ca^2+^ into the RBCs [18]. Any increase in intracellular Ca^2+^ content has been shown to promote RBC dehydration through Gardos channel activation [58]. However, the same accumulation of Ca^2+^ into RBCs during the normoxic exercise than during the hypoxic exercise was not accompanied by cellular dehydration, as shown by the lack of change in MCHC and RBC deformability. Further studies are needed to better understand why the accumulation of Ca2^+^ during exercise did not cause a change in RBC deformability, particularly during normoxic conditions. 

While the RBC rheological properties remained rather unchanged in response to the exercise performed in normoxia or hypoxia, the accumulation of Ca^2+^ and ROS into RBCs resulted in an increase in PS externalization on the RBC outer membrane leaflet in the two environmental conditions. RBC Ca^2+^ and ROS may promote the inhibition of flippase and the activation of floppase and scramblase, leading to translocation of PS to the outer membrane leaflet of RBCs [18]. In vitro studies have shown an increase in PS exposure after a 24 h incubation of RBCs in a severe hypoxic environment (FiO_2_ = 5%) [31,32]. Since hypoxic stress was significantly lower in our study, the effects of exercise may have outweighed those of hypoxia. However, our data are in agreement with another study, which showed increased percentages of PS-positive RBCs after an ultraendurance running race [20]. PS exposure is a marker of RBC senescence, which is usually accompanied by cell shrinkage. Indeed, it is surprising that the increased PS externalization was not accompanied by a major change in RBC rheological properties during exercise, both in hypoxia and normoxia. Nevertheless, we observed an activation of the coagulation after exercise in the two environmental conditions, which is in agreement with previous studies [23,59,60]. This coagulation activation cannot be explained by an increase in fibrinogen concentration since the latter was not modified by exercise in our study. However, the increased exposure of PS on the outer membrane leaflet of RBCs may have participated in the activation of coagulation. PS constitutes a privileged assembly site for the elements of the coagulation cascade and, thus, leads to thrombin generation by promoting the assembly of the tenase complex and the prothrombinase complex [21,22]. In addition, as described in the Virchow’s triad, the increase in blood viscosity after exercise could disturb blood flow by promoting venous stasis phenomena, which may also participate in the activation of coagulation [61]. Finally, although not measured in our study, the increase in the expression of some procoagulant factors after exercise, such as factors VII, VIII, IX, XII, and VWF [59,60] could also be responsible for the activation of the coagulation pathway.

## 5. Conclusions

Our study showed that acute hypoxia does not amplify the blood rheological changes, the increase in markers of red blood cell senescence, or the level of coagulation activation already induced by exercise. However, we still observed a slight decrease in RBC deformability after exercise in hypoxia only. It should be noted that the relatively small sample is a limit of our study. Therefore, additional studies will be needed to confirm this effect of hypoxia on RBC deformability and to better understand the physiological mechanisms responsible for this effect. Finally, it would be interesting to investigate the effect of different modalities of chronic training in hypoxia on blood rheology and RBC senescence. These parameters could modulate the maintenance of hematological adaptations after a return to normoxia and, therefore, performance in high level athletes.

## Figures and Tables

**Figure 1 metabolites-13-00179-f001:**
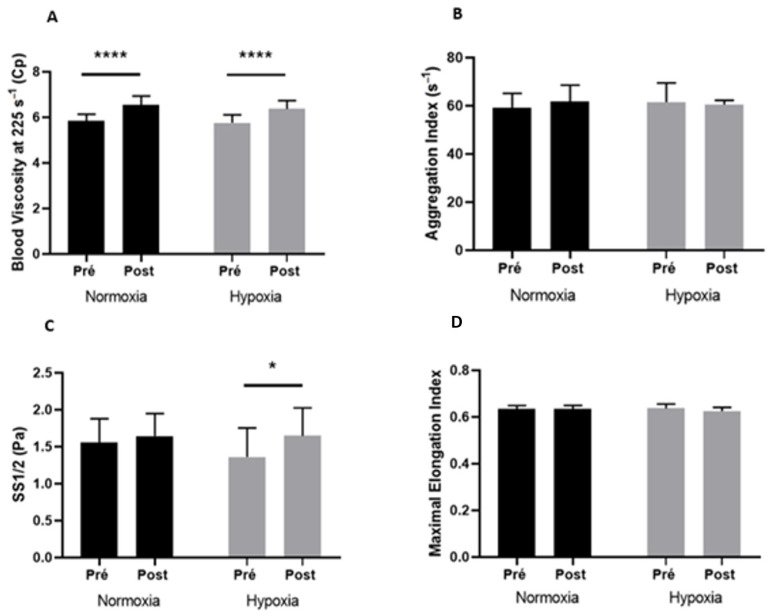
Blood viscosity at 225 s^−1^ (**A**), aggregation index (**B**), SS1/2 (**C**), and maximal elongation index (**D**) before (Pre) and after (Post) exercise, in normoxia and in hypoxia. Difference between pre- and postexercise * *p* < 0.05, **** *p* < 0.0001.

**Figure 2 metabolites-13-00179-f002:**
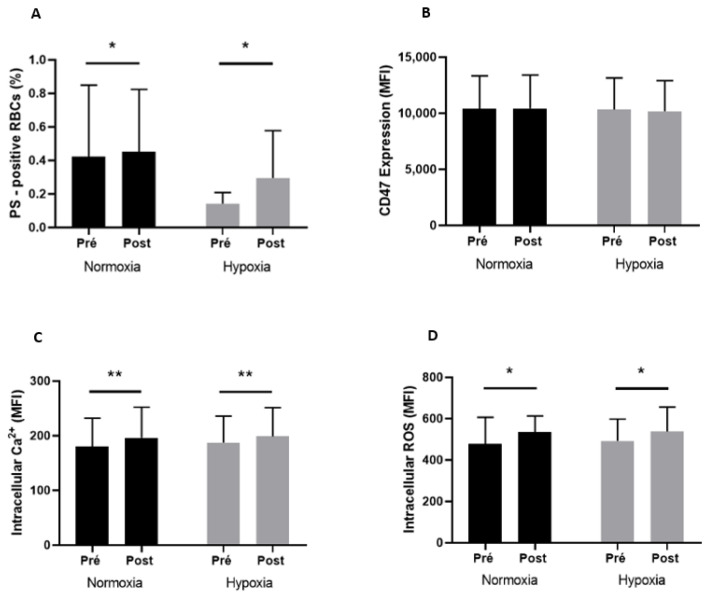
PS-positive RBCs (**A**), CD47 expression (**B**), intracellular Ca^2+^ content (**C**), and intracellular ROS content (**D**) before (Pre) and after (Post) exercise, in normoxia and in hypoxia. Difference between pre- and postexercise * *p* < 0.05, ** *p* < 0.01.

**Table 1 metabolites-13-00179-t001:** Subject characteristics during the maximal incremental test performed in normoxia and in hypoxia.

	Normoxia	Hypoxia	
VO_2max_ (mL/min/kg)	65.5 ± 9.4	56.1 ± 4.7	*
MAP (W)	408.0 ± 40.5	363.8 ± 37.4	**
VO_2_ at VT1 (mL/min/kg)	51.2 ± 9.9	40.6 ± 7.3	*
Power at VT1 (W)	297.0 ± 35.9	228.8 ± 22.3	**
VT1 (% of VO_2max_)	77.7 ± 7.8	72.1 ± 7.7	ns

VO_2max_, maximal oxygen consumption; MAP, maximal aerobic power; VT1, ventilatory threshold 1. Difference between normoxia and hypoxia * *p* < 0.05, ** *p* < 0.01.

**Table 2 metabolites-13-00179-t002:** SpO_2_, fibrinogen, blood lactate and glucose concentration, and weight.

	Normoxia		Hypoxia	
	Pre	Post	Pre	Post
Fibrinogen (g/L)	2.5 ± 0.4	2.6 ± 0.4	2.4 ± 0.2	2.6 ± 0.4
Blood Lactate (mM)	2.2 ± 0.7	5.3 ± 1.4 **	1.7 ± 0.3	5.6 ± 2.0 **
Blood Glucose (g/L)	5.2 ± 0.5	6.0 ± 1.2	4.9 ± 0.5	5.4 ± 0.4
Weight (kg)	66.2 ± 7.6	65.1 ± 7.3 ****	66.2 ± 7.5	65.2 ± 7.4 ****

SpO_2_, hemoglobin saturation. Difference between pre- and postexercise ** *p* < 0.01, **** *p* < 0.0001.

**Table 3 metabolites-13-00179-t003:** Hematological parameters.

	Normoxia		Hypoxia	
	Pre	Post	Pre	Post
Hct (%)	44.9 ± 1.8	46.6 ± 1.7 *	45,4 ± 2,3	46.6 ± 1.9 *
RBC (10^12^/L)	4.4 ± 0.2	4.6 ± 0.2 **	4.5 ± 0.2	4.6 ± 0.2 **
Hb (g/L)	133.3 ± 5.1	139.4 ± 5.5 **	135.7 ± 5.9	140.3 ± 5.6 **
WBC (10^9^/L)	4.5 ± 0.7	5.8 ± 0.9 **	4.8 ± 1.6	6.5 ± 2.1 **
PLT (10^9^/L)	146.6 ± 28.8	180.5 ± 53.7 **	158.3 ± 24.9	204.1 ± 35.2 **
MCV (fL)	88.4 ± 2.3	88.0 ± 2.4	89.3 ± 1.7	88.6 ± 2.8
MCH (pg)	30.2 ± 1.0	30.0 ± 0.8	30.3 ± 0.6	30.4 ± 0.9
MCHC (g/L)	342.1 ± 10.5	340.1 ± 8.5	339.3 ± 7.3	343.1 ± 7.8 **

Hct, hematocrit; RBC, red blood cell; Hb, hemoglobin; WBC, white blood cell; PLT, platelet; MCV, mean corpuscular volume; MCH, mean corpuscular hemoglobin; MCHC, mean corpuscular hemoglobin concentration. Difference between pre- and postexercise * *p* < 0.05, ** *p* < 0.01.

**Table 4 metabolites-13-00179-t004:** Rotational thromboelastometry parameters in NATEM and EXTEM mode.

	Normoxia		Hypoxia	
	Pre	Post	Pre	Post
NATEM Mode				
CT (s)	385.0 ± 88.9	347.8 ± 68.1	385.0 ± 59.5	344.8 ± 63.9
A5 (mm)	33.67 ± 4.8	34.11 ± 4.8	34.78 ± 2.9	34.67 ± 3.9
A10 (mm)	45.22 ± 4.6	46.56 ± 4.9	46.67 ± 2.6	47.33 ± 3.1
A20 (mm)	52.11 ± 4.5	53.89 ± 4.9 *	53.78 ± 2.7	55.0 ± 3.3 *
A30 (mm)	52.44 ± 4.6	54.33 ± 5.1 **	54.11 ± 3.3	55.67 ± 3.6 **
α (°)	62.22 ± 5.3	62.11 ± 5.7	63.56 ± 4.6	62.11 ± 5.6
CFT (s)	146.3 ± 34.0	149.7 ± 42.0	137.1 ± 26.3	147.8 ± 37.5
MCF (mm)	52.78 ± 4.6	54.56 ± 4.9 **	54.22 ± 2.7	55.89 ± 3.3 **
EXTEM Mode				
CT (s)	65.50 ± 11.0	68.88 ± 8.0	67.75 ± 6.0	67.75 ± 9.1
A5 (mm)	36.63 ± 5.5	38.50 ± 6.2	38.25 ± 4.2	38.88 ± 4.5
A10 (mm)	46.50 ± 5.4	48.63 ± 5.3 *	48.13 ± 4.2	49.13 ± 4.6 *
A20 (mm)	53.38 ± 5.4	55.75 ± 4.9 **	54.50 ± 4.2	56.25 ± 4.4 **
A30 (mm)	54.79 ± 5.2	57.04 ± 5.0 **	54.88 ± 4.2	57.25 ± 4.6 **
α (°)	67.63 ± 5.7	67.88 ± 5.2	68.88 ± 3.0	68.25 ± 3.1
CFT (s)	122.1 ± 30.0	113.0 ± 31.4	106.6 ± 16.1	109.3 ± 16.4
MCF (mm)	54.50 ± 5.0	56.63 ± 4.7	57.13 ± 5.2	57.38 ± 4.4

CT, coagulation time; CFT, clot formation time; A5, A10, A20, A30, amplitude of clot firmness 5, 10, 20, and 30 min after CT; MCF, maximum clot firmness; α, alpha angle. Difference between pre- and postexercise * *p* < 0.05, ** *p* < 0.01.

## Data Availability

The data presented in this study are available on request from the corresponding author. The data are not publicly available due to ethical reasons.

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
