# Peer review of "Effects of a Maximal Exercise Followed by a Submaximal Exercise Performed in Normobaric Hypoxia (2500 m), on Blood Rheology, Red Blood Cell Senescence, and Coagulation in Well-Trained Cyclists"

_metabolites, 2023, doi:10.3390/metabo13020179_

Round 1

Reviewer 1 Report

Comments to the Author

The authors of this article did an admirable job on an important topic, aimed to analyze the effects of an exercise conducted in normobaric hypoxia vs normoxia on blood rheology, red blood cell senescence and blood coagulation in endurance-trained cyclists. This paper is well-organized, pertinent, and may add to the literature base of an important. However, there are several points that require further clarity;

1- It would be better if the % difference values between conditions (hypoxia vs normoxia) or between pre-and post-test are given. Likewise, you can show % differences in result section (Pages 1, Lines 27-3).

2- In this section (discussion), you can give some more space to studies that examine different hypoxia conditions from different branches. You also need to further interpret the relationship of your results to exercise. This advice will help engage readers in the sports science field.

GENERAL COMMENTS:

1. The topic is important but especially the introduction and discussion sections should be improved significantly. Literature review is nonadequacy.

2. Abstract should be re-edited after changes made in the article.

Reviewer 2 Report

In this study, the authors compared the effects of acute hypoxia (FiO2 = 15.3% ≈ 2500m) versus normoxia (FiO2 = 21%) on blood rheology, red blood cell (RBC) senescence, and blood coagulation during exercise in trained male cyclists. Blood samples were taken before and after exercise to analyze various hematological parameters, including blood rheology, RBC senescence markers, and blood clot viscoelastic properties. The main findings of the study were that acute hypoxia does not amplify the blood rheological changes, the increase in markers of RBC senescence, or the level of coagulation activation induced by exercise compared to normoxia. However, the study did observe a slight decrease in RBC deformability after exercise in hypoxia. Overall, the paper is a good contribution for the literature. But it should be noticed that it involves a relatively small sample and that different results may be obtained if the athletes are subjects to harder exercise conditions.

Reviewer 3 Report

Dear Authors

Overall: The topic and methodology are novel. The article could be considered for publication in your journal after Minor revisions.

Introduction:
-The introduction is written in a structured and purposeful way. It is not necessary to mention multiple studies by mentioning the author's name in the introduction. Some of them can be transferred to the discussion.

Methods:

How is the sample size calculated?

 Please provide CV of lab tests.

Discussion:
-Discussion is
well written. But more logical reasons in favor of the hypothesis and the results should be presented.
-The results of this study have not been compared with other similar studies.

Best Regards,
